# Genome-Wide-Association-Analysis-Based Identification of Genetic Loci and Candidate Genes Associated with Cold Germination in Sweet Corn

**DOI:** 10.3390/biology14050580

**Published:** 2025-05-21

**Authors:** Changjin Wang, Yulin Yu, Jie Liu, Ahmad Rizwan, Zain Abbas, Haibing Yu, Xinxin Cheng

**Affiliations:** 1College of Agriculture, Anhui Science and Technology University, Chuzhou 233100, China; wangchj@ahstu.edu.cn (C.W.); yjs2023223@ahstu.edu.cn (Y.Y.); liujie030416@outlook.com (J.L.); ahmadrizwan4816@gmail.com (A.R.); yjs2020195@ahstu.edu.cn (Z.A.); yuhb@ahstu.edu.cn (H.Y.); 2Engineering Technology Institute of Maize Breeding in Anhui Province, Chuzhou 233100, China

**Keywords:** sweet corn (*Zea mays saccharata*), cold stress, genome-wide association study, candidate gene

## Abstract

Cold stress is a major environmental factor adversely affecting seed germination in sweet corn, resulting in reduced germination rates and yield. Sweet corn is more susceptible to cold stress than other crops, especially during germination. This study aims to identify single nucleotide polymorphisms (SNPs) and candidate genes linked to cold tolerance during germination. We evaluated phenotypic traits associated with cold germination and conducted a genome-wide association analysis (GWAS). A total of nine SNPs were found to be significantly associated with cold germination. Fourteen candidate genes related to cold germination were identified within the confidence interval. These findings provide valuable theoretical and practical insights for understanding the genetic basis of germination traits and for breeding cold-tolerant sweet corn varieties.

## 1. Introduction

Sweet corn is highly valued by consumers for its unique flavor and nutritional profile [1]. In recent years, the increasing frequency of extreme weather events because of global climate change has severely threatened sweet corn production. Compared to field corn, sweet corn exhibits significantly higher sensitivity to low temperatures [2,3]. According to statistics, global sweet corn production is reduced by 8–12% annually due to early spring low temperatures, resulting in direct economic losses of about USD 500 million [4,5]. In northern China, soil temperatures during the sowing period often drop below 15 °C. This cold stress significantly impairs the germination rate of sweet corn, disrupts proper seedling establishment, and, ultimately, leads to reduced crop yields [6,7,8]. Therefore, enhancing seed vigor and germination rates is crucial for ensuring successful seedling establishment and subsequent growth in sweet corn.

Low-temperature stress includes cold stress (15 °C) and freezing stress (<0 °C) [2]. Seed germination is a complex process that involves various metabolic reactions and signaling transduction pathways [9]. Cold stress induces physiological and biochemical alterations, including membrane rigidification, reactive oxygen species accumulation, protein denaturation, salicylic acid metabolism, and antioxidant enzyme activities [10]. Additionally, cold stress can adversely affect photosynthetic mechanisms and disrupt normal plant metabolism [11]. Although seed pretreatment with chemicals, such as salicylic acid, melatonin, and chitosan, can mitigate low-temperature damage to maize seed germination, it also significantly increases seed costs [12,13]. Therefore, genetic improvement represents a potentially effective strategy to enhance cold tolerance in sweet corn without additional costs.

The rapid development of high-throughput sequencing technologies has greatly improved the ability of genome-wide association studies (GWAS) to elucidate the genetic basis of quantitative traits and identify loci underlying complex traits [14]. Compared to traditional linkage analysis, GWAS demonstrates significant advantages in the breadth of study subjects, mapping accuracy, analytical throughput, cost efficiency, and result reliability [15]. Through association mapping, researchers have successfully identified multiple quantitative trait loci (QTLs) and candidate genes associated with cold germination in major crops, including rice [16], wheat [17], and maize [18,19]. Wang et al. [20] performed a GWAS on 295 rice germplasm lines under cold stress, identifying 67 QTLs linked to seedling cold tolerance. Through association analysis, four SNPs and 12 QTLs associated with cold tolerance were identified in maize seedlings, along with the discovery of a cold tolerance gene [21]. Wu et al. [2] identified 16 loci linked to cold tolerance in sweet corn through association mapping, along with six candidate genes associated with cold tolerance. The association mapping revealed 15 SNPs and four cold tolerance genes associated with seed germination under cold stress [7]. Zhao et al. [22] identified 76 SNP markers and 85 candidate genes linked to cold tolerance in wheat seedlings.

Despite identifying several QTLs and genes associated with cold germination in maize, studies focusing on sweet corn are still scarce, and the genetic mechanisms behind this trait remain unclear. This study used 100 sweet corn micro-core germplasms (inbred lines) as the association panel to evaluate two key germination traits under cold stress and conducted GWAS using the BLINK model. The objective of our research was to identify candidate genes associated with cold tolerance in sweet corn. These findings enhance our understanding of the genetic basis underlying cold germination in sweet corn, providing a solid theoretical foundation for the development of cold tolerance varieties.

## 2. Materials and Methods

### 2.1. Plant Materials and Phenotype Collection

In this study, 100 micro-core germplasms (inbred lines) from the Engineering Technology Institute of Maize Breeding in Anhui Province (Chuzhou Fengyang, China) were selected as experimental materials. These germplasms were collected from breeding programs in southeastern China and include tropical, subtropical, and temperate materials [23]. They have different genetic backgrounds and exhibit significant genetic variation in phenotypic traits related to tolerance to abiotic stress. Detailed information on the sources of these materials is provided in Appendix A. Before conducting the germination test, seed moisture content (MC) was calculated using the weight difference method. The dry seed weight was recorded as the initial seed weight (ISW). After weighing, the seeds were disinfected with 5% sodium hypochlorite for 10 min and rinsed four times with sterile water for the germination test. The seeds were evenly placed on the germination paper and germinated in the dark for 7 days under cold conditions at 15 °C (60% relative humidity) [24]. The moisture of the germination paper was maintained by regularly using distilled water. The experiment was performed with three biological replicates, each consisting of 50 healthy sweet corn seeds. After 7 days of germination, the seedlings and seeds were separated and dried in an oven at 105 ± 1 °C until a constant weight was achieved. The sample weights were recorded separately as seedling dry weight (SDW; g/seed) and residual seed dry weight (RSDW). The initial seed dry weight (ISDW), the weight of the mobilized seed reserve (WMSR; g/seed), and seed reserve utilization efficiency (SRUE; g/g) were calculated as follows [25]:
ISDW = ISW × (1 − MC);
WMSR = ISDW − RSDW;
SRUE = SDW/WMSR.

### 2.2. Phenotype Assessment and Processing

The phenotypic data for SRUE and WMSR were processed using Microsoft Excel 2019. Descriptive statistics, such as the range, average, standard deviation (SD), and coefficient of variation (CV), were calculated using IBM SPSS Statistics 24.0. The correlation coefficient was calculated, and the frequency distribution was visualized using the ggplot2 package. Heritability was calculated using the formula *h*^2^ = σ^2^_g_/(σ^2^_g_ + σ^2^_e/n_), where σ^2^_g_ and σ^2^_e_ represent the genetic variance and residual variance, respectively, and n indicates the number of experimental replicates [26].

### 2.3. Genome-Wide Association Study

The present study was primarily based on previous research on population structure and included the following analyses [27]. During the genotype data quality control process, we excluded low-quality SNP markers with a missing rate > 20% and a minor allele frequency (MAF) < 0.05. A total of 36,747 high-quality SNPs were retained for subsequent association analysis. To account for population structure and kinship, the BLINK model was employed as the most appropriate approach for GWAS. Principal Component Analysis (PCA) and the kinship matrix (K) were included as covariates to control for false positives and false negatives. The genome-wide significance threshold for SNP identification was set at −log10(P) > 3.6 (*p* < 1.8 × 10^−4^) based on Bonferroni correction. Significant correlations were evaluated using Q-Q plots and Manhattan plots. The LD decay of the population was calculated using PopLDdecay 3.42 software with a decay distance of 200 kb [23].

### 2.4. Functional Annotation of Candidate Genes

The 200 kb upstream and downstream regions were used to identify candidate genes [28]. Genetic loci and gene annotation information were obtained from the B73 RefGen_v3 reference genome in the MaizeGDB database (https://maizegdb.org/; accessed on 4 December 2024). Functional annotation and collection of biological information for candidate genes were conducted using both the UniProt (http://www.uniprot.org/; accessed on 7 December 2024) and NCBI databases (http://www.ncbi.nlm.nih.gov/; accessed on 7 December 2024). GO and KEGG enrichment analyses of candidate genes were conducted using the OmicShare cloud platform (https://www.omicshare.com/; accessed on 15 December 2024) [29]. Gene expression data at different developmental stages of maize were obtained from the Maize GDB database. To minimize the noise interference from lowly expressed genes, FPKM ≥ 1 was used as the threshold for selecting candidate genes [30]. Gene expression levels were normalized using log_2_ (FPKM+1) [31]. Heatmaps were employed to assess the differential expression of candidate genes across various tissues and developmental stages. Protein–protein interaction networks for the candidate genes were analyzed using the STRING v12.0 data analysis platform (https://cn.string-db.org/; accessed on 18 December 2024), and the network was visualized using Cytoscape v3.10.2 software [32].

### 2.5. Linkage Disequilibrium and Allele Effect Analysis

Genotype data covering 200 kb flanking regions of significant SNPs were extracted for haplotype analysis. An r^2^ ≥ 0.8 threshold was used to define LD blocks, ensuring a high degree of linkage disequilibrium among the SNPs within the blocks [33]. The linkage disequilibrium heatmap was visualized using the R package LDheatmap (version 1.0-6). The allelic effects of significant SNPs were evaluated by integrating corresponding phenotypic and genotypic datasets, and the results were visualized using the ggplot2 package.

## 3. Results

### 3.1. Phenotypic Descriptions of Cold Germination in Sweet Corn

To elucidate the phenotypic variation characteristics of cold-germination-related traits in sweet corn, we conducted descriptive statistics and genetic parameter assessments for two germination traits. For the SRUE trait, the range of variation was 0.10 to 0.47 g/g, with a mean of 0.27 g/g, a standard deviation of 0.084, and a coefficient of variation of 31.11% (Table 1). For the WMSR trait, the phenotypic variation varied from 0.009 to 0.092 g/seed, with a mean of 0.041 and a coefficient of variation of 37.50%. Both SRUE and WMSR showed a normal distribution trend (Figure 1). Notably, the cold-tolerant line YNY9-4 exhibited the highest SRUE value (0.470), while the cold-sensitive line JT2 (M) showed the lowest (0.100). In WMSR, the extreme pair consisting of lines HT060 (0.092) and T9357 (0.009) exhibits a significant phenotypic difference (Appendix A). These differences indicate that cold stress has a varying impact on seed germination across different lines. The correlation coefficient between these two traits was low. Both traits exhibited high heritability (*h*^2^ > 90%) and significant coefficients of variation (CV > 30%), indicating abundant phenotypic variation within the study population. This variation was mainly controlled by genetic factors, indicating potential for genetic improvement and making the population suitable for further association analysis and breeding selection.

### 3.2. Genome-Wide Association Analysis

The BLINK model was used to identify genetic loci associated with cold germination traits in sweet corn. Comprehensive genome-wide association analysis was performed based on Bonferroni correction, using a threshold of *p* < 1.8 × 10^−4^ (−log_10_P > 3.6), to evaluate the two germination traits, SRUE and WMSR (Appendix A). GWAS identified nine SNPs associated with cold germination located on chromosomes 1, 4, 7, 8, and 10 (Figure 2). Specifically, five SNPs were significantly associated with SRUE, explaining 11.3% to 15.1% of the phenotypic variation (Figure 2a,b). Among them, Affx-90948621 showed the strongest correlation with SRUE, with a *p*-value of 2.57 × 10^−6^, located on chromosome 8 and explaining 12.3% of the phenotypic variation. Four SNPs showed significant WMSR associations, accounting for 9.8–17.2% of phenotypic variation (Figure 2c,d). The most significantly associated SNP for WMSR was Affx-91309206, with a *p*-value of 5.32 × 10^−6^. This locus was located on chromosome 10 and explained 15.2% of the phenotypic variation. The PVE values of these SNPs ranged from 9.8% to 17.2%. The results indicate that the cold germination trait in sweet corn was regulated by multiple genetic loci, suggesting the complexity of its genetic basis. This provides important genetic information for future molecular-marker-assisted selection and cold tolerance improvement.

### 3.3. Candidate Gene Analysis

The association analysis identified nine SNPs linked to cold germination in sweet corn, with 207 candidate genes located within the SNP confidence intervals. Among these, 63 genes had functional annotations (Appendix A). To further elucidate the functions of these genes, we conducted GO and KEGG pathway enrichment analyses. The results indicated that these genes were significantly enriched in GO terms related to stimulus response, stress response, binding, defense response, and protein binding (Figure 3, Appendix A). Additionally, they were notably enriched in pathways, including phenylpropanoid biosynthesis, endocytosis, SNARE interactions in vesicular transport, and phagosome (Figure 3). These findings suggest that these genes hold significant potential for further investigation and exploration. In particular, the genes enriched in stimulus response, stress response, and defense response may include candidate genes related to cold tolerance.

Based on functional expression profiles and GO functional analysis, we identified 14 candidate genes crucial for cold germination, along with their corresponding orthologs in Arabidopsis and rice (Table 2, Appendix A). Among them, seven genes with orthologs linked to seed germination in Arabidopsis were identified: *GRMZM2G394528* (putative methyltransferase), *AC226248*.1_*FG002* (putative receptor-like protein kinase), *GRMZM2G010348* (cytochrome c), *GRMZM2G077937* (disease resistance protein RPM1), *GRMZM2G095164* (rapid alkalinization factor 1), *GRMZM2G473138* (N-methyltransferase ATXR7), and *GRMZM5G898755* (lipid transfer protein 1). Five genes were associated with cold stress response: *GRMZM2G095114* (Sm-like protein 4), *GRMZM2G303337* (sucrose phosphate synthase), *GRMZM2G101928* (zinc-induced facilitator-like 1), *GRMZM2G387760* (Sec1/munc18-like (SM) protein), and *GRMZM2G021416* (serine/threonine-protein kinase). Additionally, *GRMZM2G146004* (senescence-associated gene 21) and *GRMZM2G304965* (pentatricopeptide repeat-containing protein) were involved in abiotic stress responses. Notably, the candidate gene *GRMZM2G394528* identified at the Affx-91316634 locus is associated with seed germination. With a high phenotypic variation explained of 15.14%, this gene significantly impacts the phenotypic variation of cold germination traits. At the Affx-90243304 locus, two stress-related genes, *GRMZM2G303337* and *GRMZM2G304965*, were identified. This locus explains 17.18% of the phenotypic variation, emphasizing the crucial role these genes play in influencing phenotypic differences under cold stress.

To further investigate the expression characteristics of the candidate genes, we analyzed their dynamic expression patterns across various developmental stages using RNA-seq data. These genes exhibited significant variation in expression across different tissues (Figure 4, Appendix A). For example, the average expression levels of *GRMZM2G304965* and *GRMZM2G387760* were relatively low during the germination stage. In contrast, *GRMZM2G095114*, *GRMZM2G095164*, *GRMZM2G146004*, and *GRMZM2G010348* exhibited significantly higher expression levels during germination. Notably, *GRMZM2G010348* (cytochrome c) exhibits high expression across different plant tissues, and it enhances cold tolerance by regulating the levels of gibberellin and DELLA proteins in a synergistic manner [34]. *GRMZM2G021416* encodes a serine/threonine protein kinase that is highly expressed in seedling leaves. Under cold induction, its transcriptional level is upregulated, enhancing plant cold tolerance by activating cold-responsive genes and regulating ROS homeostasis [35]. Additionally, the expression levels of *GRMZM5G898755* and *GRMZM2G021416* were generally low across different developmental stages, except in seedling-related tissues. *GRMZM2G303337* has not detected gene expression data. These candidate genes play a critical role in seed germination, cold response, and the response to other abiotic stresses. These findings facilitate the investigation of cold-responsive mechanisms during the germination of sweet corn.

### 3.4. Construction of the Regulatory Network for Candidate Genes

The protein–protein interaction networks of the 14 candidate genes were constructed using the STRING database (https://cn.string-db.org/, accessed on 18 December 2024), with a moderate confidence threshold (interaction score > 0.4). Notably, 12 genes were identified through the control network and interacted with various functional proteins, forming a distinct cluster involved in regulating various biological networks (Figure 5, Table 2, Appendix A). In contrast, no significant interactions were detected for the proteins encoded by *GRMZM2G394528* and *GRMZM2G387760*, with confidence scores below the threshold of 0.4. Among the genes involved in network interactions, *GRMZM2G146004* (M3) serves as a hub node in the network, interacting with five proteins. This gene encodes a senescence-associated protein that cooperates with GST antioxidant enzymes to eliminate ROS, playing a protective role in stress responses [36]. *GRMZM2G010348* (M6) was annotated as cytochrome c. It modulates the accumulation of DELLA proteins by interacting with GA, thereby influencing plant cold tolerance [37]. Also, it interacts with CYTc1 and CYTc1-1, participating in redox reactions. *GRMZM2G473138* (M10) was annotated as N-methyltransferase ATXR7, which interacts with five proteins. Its interaction with HID1 is associated with the regulation of seed germination [38]. *GRMZM2G021416* (M14) helps plants adapt to environmental stress by regulating stress responses and circadian rhythms [35]. Additionally, it interacts with APRR3 to regulate the circadian clock.

These findings indicated that cold-germination-related traits were regulated by DEELA, GA, HID1, and a range of complex regulatory factors.

### 3.5. Analysis of Linkage Disequilibrium and Allelic Variation Effects

To validate the reliability of the significant loci and investigate their genetic background, genotype data from the 200 kb upstream and downstream regions of the significant loci were extracted for linkage disequilibrium analysis. The results revealed that seven SNPs (Affx-90392808, Affx-90891906, Affx-90948621, Affx-91198795, Affx-115332496, Affx-90990901, and Affx-91309206) were located within a highly linked region (Figure 6a,c). The dominant alleles of SNPs within the linkage region were selected to evaluate the effects of these allelic variations. For SRUE, significant phenotypic differences were observed at the Affx-90891906, Affx-90392808, and Affx-91198795 loci (Figure 6b, Appendix A). Compared to the CC and AA alleles, the average SRUE increased by 0.06 to 0.07 g/g. For WMSR, the phenotypic differences between alleles were statistically significant (Figure 6d). When comparing the GG allele with the AA/TT alleles, the average increase in WMSR was recorded as 0.011 to 0.013 g/seed. Therefore, CC and GG can be considered superior alleles.

## 4. Discussion

### 4.1. Identification of QTLs for Seed Germination Traits Under Cold Stress Using an Association Panel

Cold stress is a critical environmental factor impairing maize seed germination and seedling establishment [39]. Because of changes in climatic conditions, cold stress poses a threat to sweet corn production. To achieve better economic returns, farmers typically sow seeds in early spring. However, the low temperatures in early spring require seeds to have higher germination and emergence rates to adapt to this environment. Therefore, developing cold-resistant corn varieties and improving cold tolerance through gene modification is a crucial and feasible strategy.

With continuous updates and improvements to maize reference genome sequencing, GWAS has become an important research method for revealing the genetic basis of complex traits in maize [40]. Hu et al. [6] identified 17 genetic loci and 18 cold-tolerance-related genes during seed germination through GWAS. The association analysis identified thirty SNPs associated with seed germination tolerance under low-temperature conditions [41]. Through association mapping, Li et al. [42] identified 58 SNPs and discovered 36 candidate genes linked to seed germination. Additionally, GWAS revealed 14 SNPs associated with cold germination tolerance in maize [24]. This study identified nine SNPs significantly associated with cold germination across two germination traits, located on chromosomes 1, 4, 7, 8, and 10. The PVE values of these SNP loci all exceeded 5%, indicating a dominant effect on the target trait. To elucidate the genomic basis of cold germination in sweet corn, we performed comparative mapping between the loci identified herein and previously reported genomic regions. The Affx-90891906 and Affx-90392808 SNPs identified in this study fall within the qSRUE1 QTL interval reported by Cheng et al. [43], located between 94.1 and 103.8 Mb on chromosome 1. The Affx-91198795 locus on chromosome 8 was located 0.62 to 2.7 Mb away from the previously reported ss196425965, Affx-91281675, and Affx-90828720 [6,23]. Additionally, the Affx-115332496 locus on chromosome 7 was 1.9 Mb away from AX-86274990 reported by Wu et al. [2]. On chromosome 10, the distance between Affx-91309206 and SYN5516 was 2.8 Mb [41]. This study identified five genetic loci that overlap with regions reported in previous research, indicating that cold germination traits may be regulated by some conserved genetic factors (Figure 7). However, significant differences in the identified SNP loci were also observed. These differences may arise from variations in germplasm, environmental conditions, and the traits examined. Additionally, factors like false positive rates, resolution, and model selection in GWAS, along with stress conditions and sample size in experiments, may also contribute to these discrepancies [44,45]. Future research will explore these factors to clarify the functional roles of these loci and their interactions in the cold tolerance of sweet corn.

The linkage disequilibrium analysis revealed that seven SNP loci were located within highly linked genomic regions. These linked SNPs showed significant differences in phenotypic performance. Specifically, individuals carrying the CC allele exhibited an increase of 0.06 to 0.07 g/g in SRUE compared to those with the AA allele. The GG allele significantly outperformed the AA and TT alleles in WMSR, with an increase of 0.011 to 0.013 g/seed. The results further suggest a possible association with the target traits. In the future, cold tolerance varieties can be developed by pyramidically accumulating superior alleles. These loci could contribute to understanding the biological mechanisms underlying cold germination.

### 4.2. Candidate Genes Involved in Cold Germination of Sweet Corn

In this study, 207 candidate genes were identified from nine SNPs, of which 63 had functional annotations. Functional expression analysis of these genes revealed that 14 genes were associated with cold germination, including 7 SRUE genes and 7 WMSR genes. For SRUE, *GRMZM2G095114* was located at the Affx-90948621 locus and encodes Sm-like protein 4 (LSM4). LSM4 methylation significantly contributes to improving plant resilience under abiotic stress conditions [46]. The LSM2-8 complex helps plants regulate cold tolerance by precisely splicing the mRNA of key genes in response to cold stress [47]. The Affx-91198795 locus contains two candidate genes: *AC226248.1_FG002* and *GRMZM2G010348*. *AC226248.1_FG002* encodes a putative receptor-like protein kinase, which can phosphorylate the DEELA protein, regulating the stability and activity of the DELLA protein [48]. DELLA is a crucial negative regulator of GA signaling, primarily controlling seed germination by inhibiting GA signaling [49]. GA promotes seed germination by binding to its receptor GID1, which triggers the ubiquitination and degradation of DELLA proteins, thereby alleviating their inhibitory effect on plant growth [50,51]. Thus, it can be inferred that the *AC226248.1_FG002* gene indirectly participates in GA signaling to regulate seed germination. Furthermore, the candidate gene *GRMZM2G010348* was found to encode cytochrome c (CYTc). The absence of this gene delayed seedling growth and development, increased starch and glucose accumulation, reduced GA levels, and increased DELLA protein levels [34,37,52]. Low temperature induces the accumulation of DELLA proteins, which regulate plant growth and development, enabling plants to better cope with cold stress [53]. Therefore, it is suggested that this gene is involved in regulating cold stress tolerance.

For WMSR, the gene *GRMZM2G303337* at the Affx-91309206 locus encodes sucrose phosphate synthase. This enzyme is primarily responsible for sucrose transport, and it is essential for sustaining and promoting the growth of seedlings and root apical meristems [54]. Under cold stress, overexpression of this gene modified sugar accumulation patterns, significantly increasing glucose and sucrose levels and reducing cellular damage [55,56]. These findings suggest that *GRMZM2G403337* imparts cold tolerance through enhanced sugar accumulation and attenuated cellular damage. The gene *GRMZM2G101928* at the Affx-90990901 locus was annotated as miR165a, which regulates ABA metabolism by inhibiting HD-ZIP III [57]. In stress responses, ABA is a key hormonal regulator under low-temperature conditions. Its accumulation can activate the expression of a series of cold-responsive genes, thus enhancing plant cold tolerance [58]. Additionally, overexpression of miR165a upregulates zinc/iron transporter genes, maintaining ion homeostasis and reducing oxidative damage under low temperatures [59]. The gene *GRMZM2G021416* at the Affx-115332496 locus is homologous to OsWNK1 in rice. OsWNK1 exhibits differential expression under various abiotic stresses, with its transcript level significantly upregulated under cold stress [35]. This gene regulates endogenous rhythms and is essential for abiotic stress tolerance [60]. In summary, these candidate genes play a crucial role in seed germination, cold tolerance, and response to abiotic stresses.

## 5. Conclusions

In this study, nine SNPs were identified as significantly associated with cold germination. The confidence interval contained 207 candidate genes, 63 of which were functionally annotated. Fourteen candidate genes involved in cold germination were identified based on a literature analysis and functional characterization. These candidate genes are mainly involved in seed germination, cold tolerance, and responses to other abiotic stress. Because the functional prediction of candidate genes mainly relies on bioinformatics analysis and phenotypic traits are influenced by environmental factors, further evaluation in multi-environment trials is necessary. Future research will focus on the functional validation of candidate genes and the development of molecular markers for significant SNPs. This study advances our understanding of the genetic and molecular mechanisms underlying cold germination, establishing a theoretical foundation for breeding cold-tolerant sweet corn varieties.

## Figures and Tables

**Figure 1 biology-14-00580-f001:**
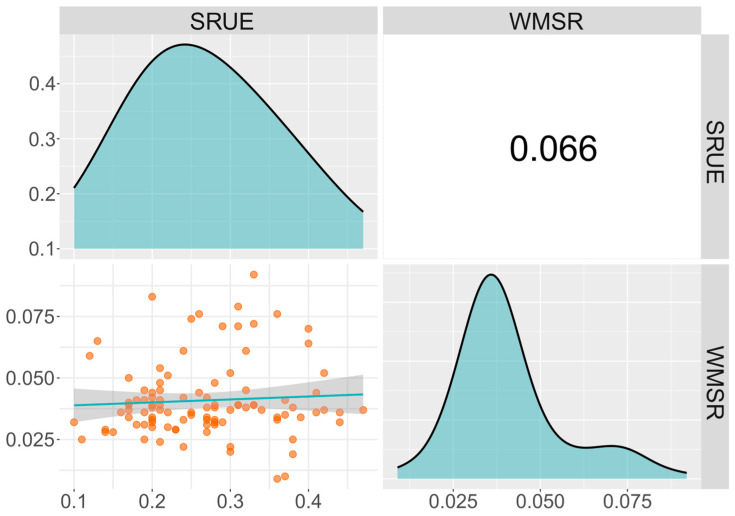
The phenotypic frequency distribution and the correlation between the two germination traits. SRUE: seed reserve utilization efficiency (g/g); WMSR: weight of mobilized seed reserve (g/seed). The phenotypic frequency distributions of each trait are shown above the diagonal. The scatter plots and correlation coefficients between traits are illustrated in the regions below and above the diagonal. The blue line in the scatter plots represents the correlation trend.

**Figure 2 biology-14-00580-f002:**
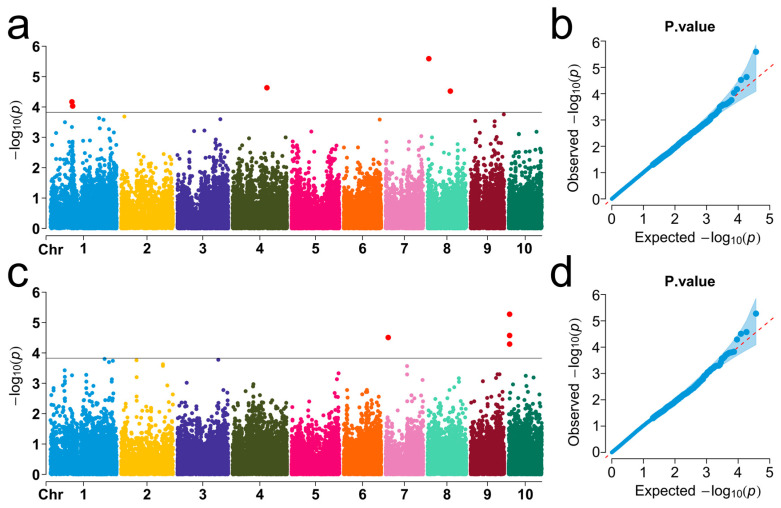
Manhattan plots (left) and Q-Q plots (right) display significant SNPs associated with SRUE and WMSR. (**a**,**b**) The Manhattan plot and the Q-Q plot for SRUE, respectively. (**c**,**d**) The Manhattan plot and the Q-Q plot for WMSR, respectively. In the Manhattan plots, the X-axis indicates chromosomal positions, while the Y-axis indicates the −log_10_ (*p*-values) for each marker. The black horizontal line indicates the genome-wide significance threshold. Red dots indicate significant SNPs exceeding the threshold. In the Q-Q plots, the red diagonal line corresponds to the expected theoretical distribution.

**Figure 3 biology-14-00580-f003:**
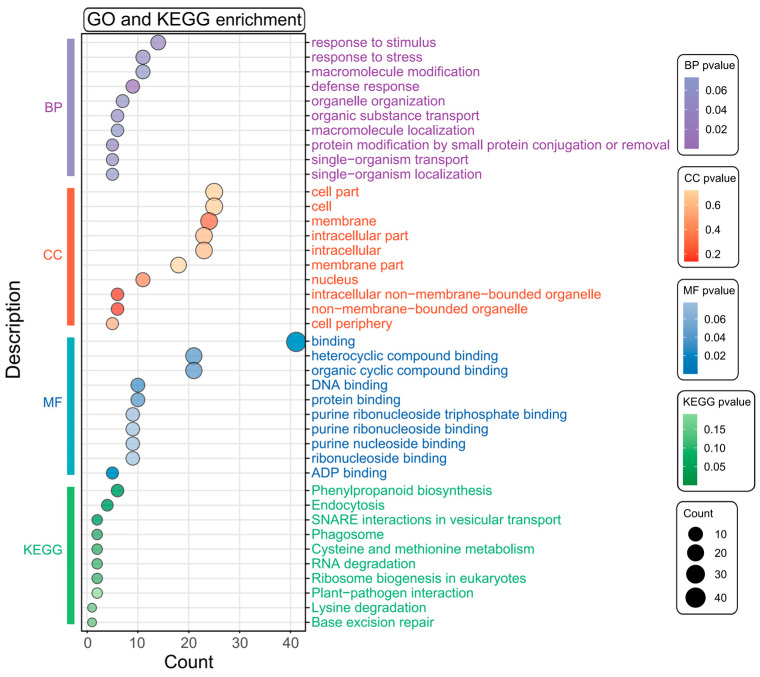
The GO and KEGG enrichment analysis of candidate genes. The vertical bar on the left shows significantly enriched GO terms (BP: biological process; CC: cellular component; MF: molecular function) and KEGG pathways. The X-axis represents the number of genes, and the Y-axis represents the enriched terms. The size of the points and the color scale represent the number of enriched genes and the level of enrichment significance, respectively.

**Figure 4 biology-14-00580-f004:**
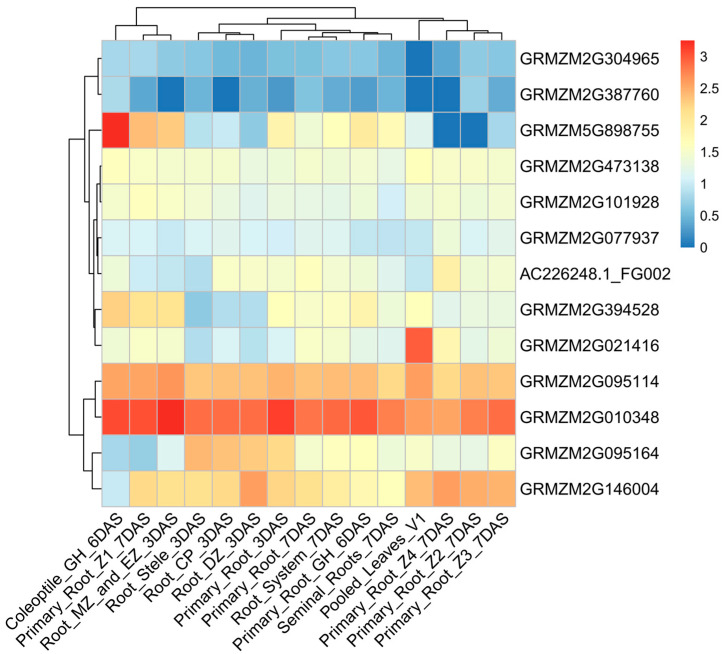
Dynamic expression characteristics of candidate genes. The scale represents the relative expression level of the gene. The X-axis represents different developmental stages and tissues, while the Y-axis represents the candidate genes. The gradient from blue to red represents expression levels from low to high.

**Figure 5 biology-14-00580-f005:**
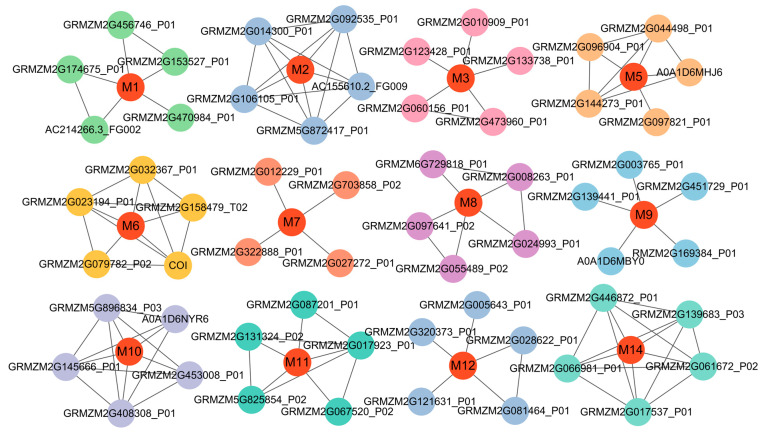
The protein–protein interaction network among different candidate genes. The black lines represent the functional interactions between proteins. M1 to M14 represent the proteins encoded by the candidate genes. The circles represent proteins. Red nodes represent hub proteins with multiple interactions.

**Figure 6 biology-14-00580-f006:**
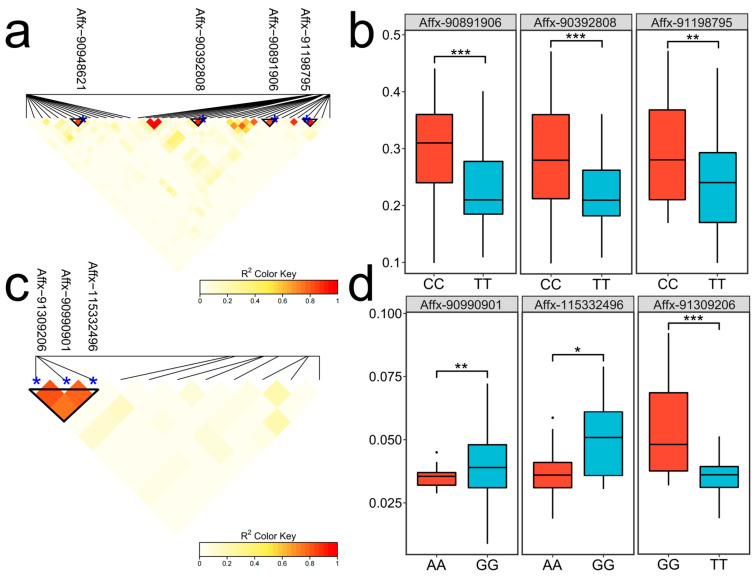
Linkage disequilibrium heatmap and allele effects of SNPs. (**a**,**c**) Linkage disequilibrium heatmaps for SNPs associated with SRUE and WMSR, respectively. The triangular boxes indicate highly linked blocks. The blue dots indicate the positions of the SNPs. (**b**,**d**) The SNP allele effects of SRUE and WMSR, respectively; * *p* < 0.05; ** *p* < 0.01; *** *p* < 0.001.

**Figure 7 biology-14-00580-f007:**
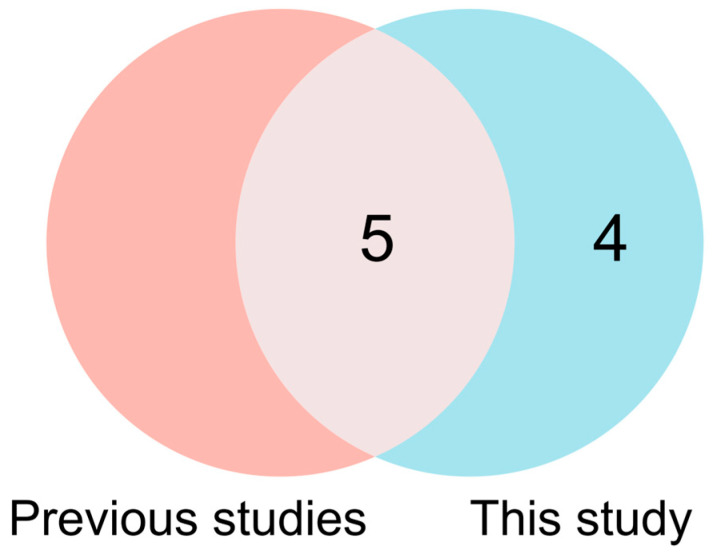
Venn diagram showing the overlap of SNPs between previous studies and the current research. Red represents the results from previous studies, while blue indicates the results identified in this study.

**Table 1 biology-14-00580-t001:** Phenotypic performance of cold germination in sweet corn.

Trait	Average ± SD	Skewness	Kurtosis	Range	CV (%)	*h* ^2^
SRUE	0.27 ± 0.084	0.315	−0.642	0.10–0.47	31.1	93.1
WMSR	0.04 ± 0.015	1.162	1.440	0.009–0.092	37.5	91.4

SRUE: seed reserve utilization efficiency (g/g); WMSR: weight of mobilized seed reserve (g/seed); SD: standard deviation; CV: coefficient of variation; *h*^2^: heritability.

**Table 2 biology-14-00580-t002:** Candidate genes associated with cold germination and gene annotations.

Trait	SNP	Chr.	PVE	Candidate Gene	Gene Annotation	Module
SRUE	Affx-90948621	8	12.26%	GRMZM2G095164	Rapid alkalinization factor 1	M1
				GRMZM2G095114	Sm-like protein 4	M2
				GRMZM2G146004	Senescence-associated gene 21	M3
	Affx-91316634	4	15.14%	GRMZM2G394528	Putative methyltransferase	/
	Affx-91198795	8	11.25%	AC226248.1_FG002	Putative receptor-like protein kinase	M5
				GRMZM2G010348	Cytochrome c-like 2	M6
	Affx-90392808	1	12.32%	GRMZM2G077937	Disease resistance protein RPM1	M7
WMSR	Affx-90243304	10	17.18%	GRMZM2G303337	Sucrose phosphate synthase	M8
				GRMZM2G304965	Pentatricopeptide repeat-containing protein	M9
	Affx-90990901	10	11.78%	GRMZM2G473138	N-methyltransferase ATXR7	M10
				GRMZM2G101928	Zinc induced facilitator-like 1	M11
	Affx-91309206	10	15.24%	GRMZM5G898755	Lipid transfer protein 1	M12
				GRMZM2G387760	Sec1/munc18-like (SM) protein	/
	Affx-115332496	7	9.82%	GRMZM2G021416	Serine/threonine-protein kinase	M14

Chr.: chromosome. SRUE: seed reserve utilization efficiency. WMSR: weight of mobilized seed reserve. PVE: phenotypic variation explained. The module indicates the protein–protein interaction networks encoded by the candidate genes. M1 to M14 represent proteins that form significant protein interaction networks in the STRING database (interaction score > 0.4), while ‘/’ indicates proteins that do not form significant interaction networks (interaction score < 0.4).

## Data Availability

The original data can be found in this paper and Appendix A.

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
