# Peer review of "Genome-Wide-Association-Analysis-Based Identification of Genetic Loci and Candidate Genes Associated with Cold Germination in Sweet Corn"

_biology, 2025, doi:10.3390/biology14050580_

Round 1

Reviewer 1 Report

Comments and Suggestions for Authors

Introduction

It is recommended to include relevant statistics or case studies that illustrate the significant economic impact of low-temperature stress on sweet corn crops, in order to emphasize the practical importance and urgency of the proposed research.

To strengthen the motivation of the study, it would be beneficial to add a sentence highlighting the genetic specificity of sweet corn or the differences in cold germination traits compared to common maize.

Materials and Methods

Where is Supplementary Table S1 and S2 uploaded or made available? Could you please clarify where it can be accessed?

2.4. The specification of 'FPKM ≥ 1' as a threshold for selecting candidate genes based on expression levels could benefit from a more detailed justification, explaining why this particular cutoff is considered appropriate or biologically relevant.

2.5. The threshold of r² > 0.8 for defining LD blocks is a commonly used standard; however, it would be helpful to mention whether this threshold was specifically validated for the studied population or if it follows a generally accepted convention.

Results

3.1. Phenotypic descriptions

The phenotypic description could be enhanced by commenting on any extreme values or anomalies observed in the data distribution. Highlighting 1-2 specific examples of lines with exceptionally good or poor performance would make the range of variation more tangible and biologically interesting for the reader.

3.2. Genome-Wide Association Analysis

The rationale for selecting the significance threshold of is not clearly explained. Please add a brief sentence or two clarifying how this threshold was determined (e.g., based on a Bonferroni correction, FDR control, or other methodology).

3.3. Candidate Gene Analysis

It would be helpful to elaborate on what a given PVE value implies in terms of the functional contribution or importance of that SNP/locus.

In Table 2, specifically in the "Module" column (which appears to be linked to Figure 5, presumably illustrating network modules), it is noted that two genes, GRMZM2G394528 and GRMZM2G387760, are listed but do not have a module assigned. This is slightly unclear for the reader. Could you please clarify why these genes are listed in the candidate table but lack a module assignment?

This section effectively identifies candidate genes based on bioinformatic analysis. However, it currently lacks experimental functional validation (such as RT-qPCR or other targeted experiments) to support the predicted roles of these candidates. While comprehensive validation might be beyond the scope of this study, it is recommended to acknowledge this and potentially suggest that functional validations are necessary future steps.

3.4. Regulatory Network Construction

The regulatory network analysis using STRING provides a basic overview, but appears somewhat superficial as it doesn't detail or analyze specific network parameters (e.g., node degree, betweenness centrality, modularity). A deeper analysis focusing on identifying and discussing key nodes (e.g., highly connected genes or potential hub genes) would significantly strengthen this part of the study and provide more insightful biological context.

Discussion

Consider including a Venn diagram to visually highlight the overlap between the genes identified in this study and those reported in other relevant maize studies. This would help contextualize the findings within existing literature. While the functional annotation and potential roles of the candidate genes are well-discussed based on existing literature, it would significantly enhance the study's contribution to explicitly compare the specific genomic loci identified as significant in this sweet corn GWAS with loci reported in other relevant GWAS studies on maize.

Reviewer 2 Report

Comments and Suggestions for Authors

Major:

I would like to emphasize that Biology is a journal known for its high Impact Factor. Biology publishes a wide range of papers on various topics. Therefore, the main topic of the paper should be clear to all biologists. But this manuscript (MS) is oversaturated with bioinformatic data. When reading, the object of research is lost. The authors need to rewrite the work and make it clearer. E.g. some remarks:

1) Line 84-85: “Detailed information on the sources of these materials was provided in Supplementary Table S1.”, but I did not find this Supplementary Table S1. Also, the basic information about used sweet corn authors should include in the main text.

2) In continuation of the first remark, the authors need to explain in detail how they obtained the basic information for bioinformatic analysis to search for genes resistant to cold in corn. Resistant cultivars? An experiment in cold conditions?

3) Improve all legends for figures, e.g. explain 1,2,3,4,5,6,7,8,9,10 in Fig. 2.

4) Increase the font size in the Fig. 2.

5) Table 2:

  1. a) “Candidate gene” - What are the numbers listed? it is necessary to provide a link to the database used
  2. b) “Module” – I do not understand what does it mean.
  3. c) 8 SNP, but in Abstract 9 SNP?

6) Fig. 4. – explain method of the gene expression calculations, also I do not understand 2 scales.

Minor:

7) Line 12: explain SNP abbreviation.

8) Include in “Keywords” the Latin name of sweet corn.

Comments on the Quality of English Language

The English could be improved to more clearly express the research

Reviewer 3 Report

Comments and Suggestions for Authors

ABSTRACT

- The scientific objective is not clearly formulated. The study claims to have "investigated phenotypes," but it does not explicitly state a hypothesis or research question. It is unclear whether this was a purely exploratory study or if there was an intention to test an already presumed relationship between certain genetic markers and cold germination.

- Mentioning the BLINK model is welcome, but it would be useful to briefly explain why it was chosen (e.g., "for its accuracy in reducing type I errors," etc.). Otherwise, it seems like an arbitrary choice.

- It is stated that 14 genes are related to cold germination, but it is unclear by what criteria (GO functions? differential expression? orthology?). The summary could briefly mention the methodology used for selection.

- Although space is limited, it would be useful to mention at least the sample size.

INTRODUCTION

- The text jumps quite quickly from general statements about maize and abiotic stress to molecular details (genes, proteins, signaling pathways) without a clear transition. A more rigorous introduction should follow the classic structure "from general to specific."

- The statement "cold stress affects germination and seedling growth" is repeated twice, and further below, the same ideas are reiterated in a fragmented way. The information should be synthesized logically, not repeated in similar blocks.

- Examples from rice, wheat, and maize in general are provided, but it is unclear whether these conclusions can be extrapolated to sweet corn, which has genetic peculiarities (e.g., su1, sh2 loci). It is only generically stated that "studies on sweet corn are rare," but it is not shown what has been done so far on sweet corn.

- The presentation of molecular mechanisms is too detailed for an introduction. Passages about ICE1, MAPK, RAB, LTI, etc., would be more appropriate in a discussion or separate molecular background chapter, not in a 1-2 page introduction. Additionally, these seem to be copied from model species (Arabidopsis) without showing their direct relevance to sweet corn. It gives the impression of an unrelated "review."

- Mentions of specific environmental factors are missing. If cold stress occurs in the context of early spring sowing, a brief reference to actual climatic conditions would be useful (e.g., "in northern China, soil temperature at sowing often drops below 15°C").

- The objective appears in a very long sentence without a clear hypothesis. Ideally, the end of the introduction should explicitly formulate the goal, method, and what new contribution the study makes.

MATERIALS AND METHODS

- It is mentioned that the seeds were kept "7 days in the dark at 15°C," but the relative humidity, substrate type, ventilation, or possible contamination are not specified. Including these parameters is important for the reproducibility of the experiment.

- The explanations of SNP filtering, BLINK, PCA, and the kinship matrix are correct but feel generic, as if extracted from a manual rather than part of a real experiment. Justify the choice of the BLINK model over others (e.g., MLM or FarmCPU) given the population structure.

- MaizeGDB, UniProt, NCBI, etc., are mentioned, but it is not explained what was taken from each or why those sources were chosen.

- It is stated that Manhattan, QQ, and heatmap diagrams were used, but it is not indicated whether they are presented in the paper or what their relevance was for interpretation. Briefly explain what was obtained from each visualization and how it contributed to the final analysis.

RESULTS

- The section begins abruptly without a short introduction explaining what will be presented and in what order.

- Numerical values (means, standard deviations, coefficients of variation) are provided, but their biological significance is not discussed enough. For example, the coefficient of variation of 31.11% is presented as such, but it is not stated whether this suggests useful variability for selection. I suggest briefly commenting on whether high heritability values (>90%) or high CV values (>30%) support the potential for genetic improvement.

- Figures are merely described ("Manhattan and QQ diagrams...," "the red dots indicate..."), but they are not discussed in terms of biological interpretation or practical relevance. For example, it is not enough to say that an SNP has a low p-value; the biological significance of that association should also be mentioned.

- Make an explicit connection between gene expression and its relevance to the observed phenotype! Although "dynamic expression models" and variations in tissues are mentioned, no clear conclusions are drawn regarding the role of these genes in the cold response context.

- After each subsection (e.g., after 3.2, 3.3), a short paragraph summarizing the main conclusion would be useful, stating "These results show that... and suggest that...". Integrate the results periodically to help the reader avoid getting lost in technical details.

DISCUSSIONS

- The text is limited to reiterating the results and placing them in the context of other studies without a deeper comparative analysis. Differences between the cited studies and the presented study are not discussed, and similarities are presented descriptively rather than interpretively.

- No limitations are mentioned, either related to the GWAS methodology (e.g., false positive rate, resolution) or the experimental context (e.g., stress conditions applied, sample size).

- The results are presented in an excessively certain and optimistic tone ("confirming reliability," "providing valuable information"), without scientific caution. A more nuanced language would be more appropriate: "suggests a possible association," "could contribute to understanding...," "requires further validation." This is a recommended scientific practice, especially in GWAS studies.

- Some ideas are unnecessarily repeated (e.g., identification of the 9 SNPs and their connection to the target traits appears twice), which dilutes the clarity and conciseness of the chapter.

- It is mentioned that SNPs affect phenotypic performance, but no details are provided regarding their concrete impact on measurable parameters (e.g., germination rate, emergence time, seedling vigor).

CONCLUSIONS

- The phrasing "offers valuable perspectives" is vague and does not specify what can be done further based on these results.

- The conclusion does not mention any limitations of the study. A brief mention that functional validation of candidate genes is needed should be included.

- No future research direction is suggested, even though the identification of SNPs and candidate genes is well-suited for such recommendations.

Round 2

Reviewer 2 Report

Comments and Suggestions for Authors

1) “Comments 2: In continuation of the first remark, the authors need to explain in detail how they obtained the basic information for bioinformatic analysis to search for genes resistant to cold in corn. Resistant cultivars? An experiment in cold conditions?

Response 2: …

The materials used in our study include cold-tolerant varieties observed in previous field data. This experiment was conducted under cold stress conditions, as described in lines 99-100 of the text.”

- I did not find detailed information about “observed in previous field data”.

2) “Comments 5: Table 2:

b)“Module” – I do not understand what does it mean.

Response 5: The '–' listed in the 'Module' section represents proteins that form the interaction network. Since we used a medium interaction confidence threshold (>0.4), the two candidate genes listed (GRMZM2G394528 and GRMZM2G387760) have an interaction confidence lower than 0.4, and therefore, no significant interaction network was formed. We have added supplementary information in lines 282-284 and 287-289 of the manuscript.”

- explain M1 – M14 and / - in “Module” section.

Comments on the Quality of English Language

The English could be improved to more clearly express the research

Reviewer 3 Report

Comments and Suggestions for Authors

Congratulations for all your research work and for writing this paper!

Author Response

Dear reviewers and editors,

Thank you very much for taking the time to review our manuscript and for providing valuable insights. Your detailed suggestions have greatly improved the quality of the paper. We sincerely appreciate your professionalism and patient guidance throughout the review process. Your feedback not only helped improve this paper but has also been highly beneficial to us. Thank you very much.

Sincerely,

Round 3

Reviewer 2 Report

Comments and Suggestions for Authors

- I did not find the answers for those two remarks:

1) “Comments 2: In continuation of the first remark, the authors need to explain in detail how they obtained the basic information for bioinformatic analysis to search for genes resistant to cold in corn. Resistant cultivars? An experiment in cold conditions?

Response 2: …

The materials used in our study include cold-tolerant varieties observed in previous field data. This experiment was conducted under cold stress conditions, as described in lines 99-100 of the text.”

- I did not find detailed information about “observed in previous field data”.

2) “Comments 5: Table 2:

b)“Module” – I do not understand what does it mean.

Response 5: The '–' listed in the 'Module' section represents proteins that form the interaction network. Since we used a medium interaction confidence threshold (>0.4), the two candidate genes listed (GRMZM2G394528 and GRMZM2G387760) have an interaction confidence lower than 0.4, and therefore, no significant interaction network was formed. We have added supplementary information in lines 282-284 and 287-289 of the manuscript.”

- explain M1 – M14 and / - in “Module” section.

Comments on the Quality of English Language

The English could be improved to more clearly express the research

Author Response

Dear Reviewer,

We sincerely apologize for the confusion caused by our oversight. What we intended to express was that the materials we used contain cold-resistant germplasm. We provided detailed responses in the second round of review. We deeply regret the confusion caused by our carelessness and hope for your understanding.

Additionally, we are truly grateful for your careful review and the detailed suggestions for revisions you provided. Thank you very much.

Sincerely,